# Isolation of Novel *Xanthomonas* Phages Infecting the Plant Pathogens *X. translucens* and *X. campestris*

**DOI:** 10.3390/v14071449

**Published:** 2022-06-30

**Authors:** Sebastian H. Erdrich, Vikas Sharma, Ulrich Schurr, Borjana Arsova, Julia Frunzke

**Affiliations:** 1Institute of Bio- and Geosciences, Department for Plant Sciences (IBG-2), Forschungszentrum Jülich, 52425 Jülich, Germany; s.erdrich@fz-juelich.de (S.H.E.); u.schurr@fz-juelich.de (U.S.); b.arsova@fz-juelich.de (B.A.); 2Institute of Bio- and Geosciences, Department for Biotechnology (IBG-1), Forschungszentrum Jülich, 52425 Jülich, Germany; v.sharma@fz-juelich.de

**Keywords:** phage isolation, phage genomics, *Xanthomonas campestris*, *Xanthomonas translucens*, *Siphoviridae*, phage biocontrol

## Abstract

The genus of *Xanthomonas* contains many well-known plant pathogens with the ability to infect some of the most important crop plants, thereby causing significant economic damage. Unfortunately, classical pest-control strategies are neither particularly efficient nor sustainable and we are, therefore, in demand of alternatives. Here, we present the isolation and characterization of seven novel phages infecting the plant-pathogenic species *Xanthomonas translucens* and *Xanthomonas campestris*. Transmission electron microscopy revealed that all phages show a siphovirion morphology. The analysis of genome sequences and plaque morphologies are in agreement with a lytic lifestyle of the phages making them suitable candidates for biocontrol. Moreover, three of the isolated phages form the new genus “*Shirevirus*”. All seven phages belong to four distinct clusters underpinning their phylogenetic diversity. Altogether, this study presents the first characterized isolates for the plant pathogen *X. translucens* and expands the number of available phages for plant biocontrol.

## 1. Introduction

Pathogenic microbes represent a major factor hampering current food production and account for an annual loss of 10% of the global production [1]. Phage-based biocontrol for the treatment of bacterial infections was already assessed in trials by Mallman and Hemsworth in 1924 to treat black rot of cabbage shortly after the discovery of phages in general [2]. However, due to their high specificity, phages were replaced by broad-range antibiotics and cheaper copper treatments in the beginning of the last century. Nowadays, the massive spread of antibiotic-resistant bacteria has fostered a global surge in the reinvestigation of phage biology and phage-based biocontrol as an alternative to antibiotic treatments [3,4,5,6].

An overview of the economically most relevant plant pathogens is given in Mansfield et al. 2012 [7]. In recent years, several attempts have been made to isolate phages infecting plant pathogens, including *Ralstonia solanacearum*, *Erwinia amylovora*, *Pseudomonas syringe* spp, *Xylella fastidiosa*, and *Xanthomonas* spp. Multiple studies were conducted on phage biocontrol of *Ralstonia solanacearum* showing suppression of plant wilting on potato and tomato [8,9,10,11,12,13]. *Erwinia amylovora*, which developed resistance to streptomycin, has led to the evaluation of phage biocontrol with promising outcomes in some cases [14,15,16]. Due to the broad spectrum of plants infected by *Pseudomonas syringae* spp., multiple biocontrol trails have been conducted in the past, but with a special effort on citrus canker disease [17,18,19,20,21,22,23]. *Xylella fastidiosa* is a major threat to olive trees in Europe, where first phages have been isolated and tested [24,25,26].

The genus of *Xanthomonas* contains multiple gram-negative plant pathogens causing devastating losses in food production in a broad variety of important food crops, from the *Poaceae* family (including rice, sugar cane, and wheat) to the *Brassicaceae* family (including cabbages, broccoli, and oil seed). In this study, we focus on two important *Xanthomonas* pathogens: (1) *Xanthomonas translucens* pv. *translucens* (Xtt), causing bacterial leaf streak in cereals, and (2) *Xanthomonas campestris* pv. *campestris* (Xcc), the major cause of black rot disease in crucifers.

*X. translucens* (Xtt) infects cereals such as barley or wheat and, thereby, poses a threat to crop production, with annual losses reported from 10% reaching up to 40% in severe cases [27]. For other *Xanthomonas* species [6], first phages have been sequenced and morphologically characterized, but in the case of Xtt no phages are available in phage collections. There is only one report from 1953 on the isolation of four phages for different *X. translucens* pathovars [28], which were not sequenced nor further characterized so far. Therefore, the isolation of lytic phages for Xtt is required for the establishment of phage-based biocontrol strategies of this important pathogen. Xtt has been shown to be transmitted from leaf to seed, emphasizing that seed decontamination treatments are not always completely successful [27]. Outbreaks have been reported from many geographical regions. Nevertheless, they prevail in warmer regions. In the context of climate-change and worldwide rising average temperatures it is likely that the impact of these plant diseases will increase. 

As a second host for phage isolation, we chose *Xanthomonas campestris* pv. *campestris* (Xcc), a vascular pathogen causing black rot of crucifers. The rod-shaped uniflagellar yellow colony forming bacteria causes V-shaped lesions in many *Brassicaceae* plants, of which *B. oleracea* (cabbage, cauliflower, and broccoli) is the economically most relevant [29,30]. Its main route of transmission is via the seeds, but it also was shown to be persistent in field soil [31]. For Xcc, phage isolates have been reported in the past (Caudovirales: FoX 1–7 [32]; phage PPDBI, phage PBR31, ϕL7, Phage Carpasina [33], XC1, Xccφ1 [34], Tubuvirales: ϕLf2, ϕLF), but there is still a great need for further phage isolates applicable for biocontrol. Of note, some of the previously isolated phages belong to the order of Tubuvirales, which are known to cause chronic infections of their bacterial host. Chronic infections typically have a fitness cost for the bacterium, but they do not kill the bacterial cell and are, therefore, not the ideal scenario for biocontrol. 

One of the most important aspects of phage biocontrol is the isolation of phages with desirable traits, including a lytic lifestyle, the absence of virulence genes, and reasonable repression of host growth. Comprehensive characterization, genome analysis, and the analysis of the host range of newly isolated phages allow us to better understand how phages target their host bacterium. Altogether, these efforts are important to expand the “toolbox” for sustainable and targeted control of plant pathogens and allows researchers to select the phages with the most desirable traits. 

Here we present the isolation, characterization, and genome analysis of seven novel *Xanthomonas* phages. Six were isolated using *Xtt* as host bacterium (Langgrundbatt 1, Langgrundbatt 2, Pfeifenkraut, Elanor, Laurilin and Mallos), and one of them (Seregon) was isolated using *Xcc*. Transmission electron microscopy and genome sequencing revealed that all seven phages show a siphovirion morphology and have a lytic lifestyle, making them suitable candidates for phage biocontrol.

## 2. Materials and Methods

### 2.1. Bacterial Strains and Growth Conditions

*Xanthomonas translucens* pv. *translucens* (DSM 18974) [27] and *Xanthomonas campestris* pv. *Campestris* [29,30,35] were used as host strains for phage isolation in this study. Cultures were grown on a nutrient agar plate and inoculated from single colonies in liquid media for overnight cultures. Nutrient broth (NB) or agar was used for culturing the bacterial strains at 30 °C.

### 2.2. Phage Isolation and Propagation

Phages were isolated from wastewater samples donated by the Forschungszentrum Jülich wastewater plant (50.902547168169825, 6.404891888790708—Jülich, Germany) as well as from soil samples (50.754354003126345, 6.366620681310555 within a 5 km radius—Eifel, Germany).

Virus particles within the samples were solubilized using phosphate-buffered saline (100 mM NaCl, 2.7 mM KCl, 10 mM Na_2_HPO_4_, 1.8 mM KH_2_PO_4_, 1 mM CaCl_2_, and 0.5 mM MgCl_2_) and incubated for 3 h on a rocking shaker (Heidolph Polymax 1040, Heidolph Instruments GmbH and Co. KG, Schwabach, Germany) at 30 rpm. Afterwards, the samples were centrifuged at 5000 *g* for 15 min to remove solid particles within the samples. The supernatants were filtered through 0.2 µM pore size membrane filters (Sarstedt; Filtropur S, PES). For each sample, an aliquot of 1 mL was mixed with 0.4% NB soft agar and 100 µL of a densely-grown overnight culture (OD600 of 1) of the host and directly plated on an NB agar plate according to a modified version of the double agar overlay [36]. Plates were stored at 30 °C for incubation overnight. 

Phage enrichment was performed in 15 mL NB. Filtered supernatant solution (1 mL) and 1 mL of an overnight culture of the host was added. To adjust the enrichment culture to 1-fold NB, appropriate amounts of 5-fold concentrated NB were added to the sample. The culture was incubated at 30 °C and at 150 rpm overnight. Afterwards, the enrichment cultures were centrifuged at 5000× *g* for 25 min to collect the supernatant, which was subsequently filtered with 0.2 µM pore size membrane filters, to remove residual bacteria. Serial dilutions of the enrichment supernatant were spotted on double agar overlay plates containing the host bacterium. Plaques were typically visible after overnight incubation. 

Purification of the phage samples was carried out by restreaking single plaques with an inoculation loop on a fresh double agar overlay at least three times. When a stable plaque morphology was observed after three restreakings, a sample was considered as a single phage isolate [36].

Harvesting of purified phage particles was performed after overnight incubation of a double agar overlay containing the purified phage. The top agar was solubilized by adding 5 mL SM buffer (100 mM NaCl, 8 mM MgSO_4_, and 100 mM Tris-HCl) and 2 h incubation on a rocking shaker. The solution was, subsequently, transferred into a falcon tube and centrifuged at 5000× *g* for 25 min to remove the residual amounts of top agar. The supernatant was filtered through 0.2 µM syringe filters and stored at 4 °C. For titer determination, a dilution series was spotted on overlay agar and the visible plaques at the highest dilution were counted.

Phage particles were either amplified on plates or in liquid cultures. For plate amplification, a double agar overlay was performed using 100 µL phage solution with a high titer (>108 Pfu/mL) added to the top agar and harvested as described above. For liquid propagation 50 mL medium was inoculated with 1 mL host overnight culture and 100 µL phage solution and incubated at 30 °C 150 rpm overnight. The cleared lysate cultures were centrifuged at 5000× *g* for 25 min to collect the supernatant, which was subsequently filtered with 0.2 µM pore size membrane filters. Subsequently, a 10% PEG enrichment according to [37] was performed to obtain very high titers.

### 2.3. Electron Microscopy Observation of Phage Virions

For electron microscopy of single phage particles, 3.5 µL purified phage suspension was fixated on a glow discharged (15 mA, 30 s) carbon coated copper grid (CF300-CU, carbon film 300 mesh copper) and stained with 2% (*w*/*v*) uranyl acetate. After air drying, the sample was analysed with a TEM Talos L120C (Thermo Scientific, Dreieich, Germany) at an acceleration of 120 kV. 

### 2.4. Phage Infection Curves

Infection was performed in microtiter plates using the BioLector^®^ microcultivation system (Beckmann Coulter GmbH, Krefeld, Germany). For cultivation, biological triplicates were conducted in 48-well FlowerPlates (Beckmann Coulter GmbH, Krefeld, Germany) at 30 °C and a constant shaking frequency of 1200 rpm. Backscatter was measured by excitation with light of a wavelength of 620 nm (filter module: λEx/λEm: 620 nm/620 nm, gain: 40) in 15 min intervals. Each well contained 1 mL host culture adjusted to an OD_600_ of 0.2 in NB and an addition of 2 mm MgCl_2_. Phages were added at a multiplicity of infection (MOI) of 1 or 0.1, respectively, and incubated for 15 min at room temperature without shaking to promote phage adsorption. Sampling was performed at the indicated time points. Subsequently, 3 µL of dilutions were spotted on NB double agar plates containing the isolation host.

### 2.5. Host Range Determination

The host range of the phages was determined on the following strains: *Xanthomonas translucens* pv. *translucens* (DSM 18974) [27] and *Xanthomonas campestris* pv. *campestris* [29,30], *Pseudomonas syringae* (DSM 50274) [38], *Pseudomoas syringae* pv *tomato* DC3000 (Pst) [39,40], *Pseudomonas fluorescence*, *Azospirillum brasilense* sp245 grown on NB and *Sinorhizobium meliloti* 1021, *Herbaspirillum seropedicae, Pseudomonas koreensis* (DSM 16610), and *Bacillus subtilis* grown on Lysogeny Broth (LB).

The host range was determined by spotting dilution series of the phage solution on bacterial lawns prepared as double agar overlays in triplicates. A species was considered as part of the host spectrum of the phage if single plaques were visible. The efficiency of plating (EOP) was calculated relative to the isolation host.

### 2.6. DNA Isolation

For isolation of phage DNA, 2 mL of phage solution was treated with 1 U/μL DNAse (Invitrogen, Carlsbad, CA, USA) to remove free DNA from the solution and incubated for 30 min at 37 °C. Afterwards, EDTA and proteinase K were added to the mixture at final concentrations of 50 mM. SDS was added to a final concentration of 1% (*w*/*v*) to remove structural proteins. The mixture was incubated at 56 °C for 1 h. Subsequently, phage DNA was separated by adding 250 μL of phenol:chloroform:isopropanol (25:24:1; *v*/*v*). The mixed solution was centrifuged at 16,000× *g* for 4 min and the upper phase containing the DNA was carefully transferred to new microcentrifuge tube. Afterwards two volumes of chilled 100% ethanol were added as well as sodium acetate to a final concentration of 0.3 M. The samples were stored for at least 1 h at −20 °C and centrifuged afterwards for 10 min at 16,000× *g*. The supernatant was discarded, and the pellet was washed with 70% ethanol. The pellet was air-dried and finally resuspended in 50 µL DNAse free water. The purified DNA was stored at 4 °C until further usage for sequencing [41].

### 2.7. DNA Sequencing and Genome Assembly

The DNA library was prepared using the NEBNext Ultra II DNA Library Prep Kit for Illumina, according to the manufacturer’s instructions, and shotgun-sequenced using the Illumina MiSeq platform with a read length of 2 × 150 bp (Illumina, San Diego, CA, USA). For each phage a subset of 100,000 reads were sampled, and a de novo assembly was performed with CLC genomics workbench 20.0.4 (QIAGEN, Hilden, Germany). Finally, contigs were manually curated and checked for coverage.

### 2.8. Gene Prediction and Functional Annotation

Open reading frames (ORFs) were predicted with PHANOTATE v.1.5.0 [42] and annotated against the custom databases (NCBI viral proteins, NCBI Refseq proteins, Kyoto Encyclopedia of Genes and Genomes (KEGG) [43], Phage Annotation Tools and Methods (PhAnToMe) (www.phantome.org), phage Virus Orthologous Groups (pVOG)) using the multiPhATE v.1.0 pipeline [44]. Additionally, all identified sequences were later curated manually using online NCBI Blast against the non-redundant (NR) database [45]. Conserved protein domains were further predicted using the batch function of NCBI Conserved Domain Database (CDD) [46] with the e-value cutoff of 0.01.

The annotated genomes were deposited on NCBI via BankIt portal under the following accession numbers: ON189042 (Langgrundblatt1), ON189043 (Langgrundblatt2), ON189044 (Pfeifenkraut), ON189045 (Elanor), ON189046 (Laurilin), ON189047 (Mallos), and ON189048 (Seregon).

Genome termini classes were determined using Phage Term [47] parameters were set by default. Phage lifestyle was predicted by the machine-learning-based program PhageAI [48] using default parameters and further confirmed by the absence of integrase genes inside the genomes.

### 2.9. Genome Comparison and Classification

Genome maps were created using the R package gggenes (version 0.4.1) with fixed length parameters.

Novel phages were classified based on complete nucleotide sequences by comparing them against known sequenced phages infecting *Xanthomonas*, including closely related members recovered from the NCBI nucleotide blast searches, resulting in a total of 97 phage genomes. The 90 known genomes were downloaded from NCBI by their unique identifier. The average nucleotide identities (ANI) were calculated by pairwise comparison of the seven novel phages to the 90 reference genomes using the Perl program ClusterGenomes (https://github.com/simroux/ClusterGenomes) with default settings (80% coverage and 95% ANI). Further, we performed a clustering analysis using the VIRIDIC tool [49]. Heatmap clustering were displayed using the R package “pheatmap v.1.0.12”. In addition, more than 2000 proteobacteria-specific complete genomic sequences based on Virus–Host DB [50] information list of accession numbers were downloaded using the python program NCBI-genome-download (https://github.com/kblin/ncbi-genome-download). This set of sequences was clustered at 95% identity into 725 clusters using ClusterGenomes (https://github.com/simroux/ClusterGenomes) with default settings (80% coverage, 95% ANI). Lastly, a representative sequence from each 725 clusters, including seven novel phages with related phages obtained from blast-based searches, was analysed using k-mer clustering phylogeny (https://bioinformaticshome.com/bioinformatics_tutorials/R/phylogeny_estimation.html).

## 3. Results

### 3.1. Phage Isolation and Virion Morphology

Seven novel phages infecting *Xanthomonas* species were isolated from wastewater samples taken at Forschungszentrum Jülich (Germany) or from soil samples from the Eifel (Germany). The phages Langgrundbatt 1, Langgrundbatt 2, Pfeifenkraut, Elanor, Laurelin, and Mallos were isolated on *Xanthomonas translucens* pv. *translucens* (DSM 18974) (Figure 1A). Langgrundblatt 1 and 2 formed clear plaques with a mean diameter of 0.6 and 0.9 mm, respectively, but with considerable variation in plaque sizes. Phage Pfeifenkraut formed homogeneous round and clear plaques with a diameter of 1.3 mm. Phages Elanor, Laurilin, and Mallos formed clear irregular plaques with average diameter of 0.3 mm, 0.45 mm, and 0.65 mm, respectively (Figure 1B). The phage Seregon was isolated using *Xanthomonas campestris* pv. *campestris* as host. Sergon’s plaques had a diameter of 0.9 mm and were turbid in appearance. They were visible only after two days of incubation (Figure 1B).

TEM analysis of purified phage particles showed that all seven phages have an icosahedral capsid with sizes ranging 55–82 nm (Appendix A) and a non-contractile tail 134–225 nm (Figure 1C). Therefore, based on their morphology the phages were classified as siphovirion phages. 

### 3.2. Infection Curves and Host Range Determination

All isolated phages suppressed growth of the host culture when applied at a multiplicity of infection (MOI) of 0.1 or 1 (Figure 2). Interestingly, in some cases, the lower MOI had a stronger inhibitory effect on the host culture. Quantification of the phage titer by spotting dilution series on a bacterial lawn allowed us to visualize the amplification dynamics of the phages over the course of the experiment (Figure 2 right panel). Amplification was in several cases more pronounced at an MOI of 0.1.

In the context of plant biocontrol, the host range of phages represents an important parameter, since phages should not target plant growth promoting bacteria. Therefore, we assessed the host-range of the seven phages by spotting them on bacterial lawns of different *Xanthomonas* species (Xtt; Xcc), plant growth promoting bacteria (*Pseudomonas fluorescence*, *Azospirillum brasilense*, *Sinorhizobium meliloti* 1021, *Herbaspirillum seropedicae*, *Pseudomonas koreenis,* and *Bacillus subtilis*) and other plant pathogenic bacteria (*Pseudomonas syringae* pv *lapsa*, *Pseudomonas syringae* pv. *tomato,* and *Agrobacterium tumefaciens*). *Xanthomonas campestris* showed a slight susceptibility to the phages Langgrundblatt1, Langgrundblatt2, Pfeifenkraut, and Mallos. Vice versa, the phage Seregon isolated on *Xcc* showed lytic activity on *X. translucens,* but also at low levels. The EOPs are listed in Appendix A. Interestingly, phage Mallos showed additional lytic activity on the plant pathogenic *P. syringae* strain DC3000, but also infected the plant growth-promoting *P. fluorescens*. This is making phage Mallos an omnilytic phage infecting distinct species and the phage with the broadest host range of the phages isolated in this study. 

### 3.3. Genome Sequencing and Genome Features 

All isolated phages were sequenced using Illumina Mi-Seq. short-read technology. Genomic features of the seven isolated phages are summarized in Table 1. Briefly, they show genome sizes from 40 to 62 kb, a GC content varies in the range of 53 to 64% and contain 56–89 annotated ORFs.

Phages Pfeifenkraut, Langgrundbatt 1, and Langgrundbatt 2 exhibit a remarkably low GC contend (53%) in contrast to their host *Xtt* (68%). Phage Laurelin differs from the other phages, in that its genome is the smallest, containing only 56 genes. Furthermore, it is the only one of the isolated phages containing 218 bp directed terminal repeats (DTRs). The genomic ends were determined using Phage Term [47]. In contrast, phages Langgrundbatt 1, Langgrundbatt 2, Pfeifenkraut, Elanor, and Mallos show a headful packing mechanism were the genome is translocated to the capsid at dedicated *pac* sites [51] resulting in variable genome ends. For the phage Seregon, cohesive ends with a length of 12 bp (5′ GGGGGCGCTGAC) were predicted. Since for plant biocontrol the phage lifestyle is of special importance, the lifestyle was predicted using PhageAI, a machine-learning tool which compares the genomes to over 20,000 publicly available phages [48]. All isolated phages were classified as virulent. This is further supported by the absence of integrase genes within the genomes.

The genome architecture for our isolated phages was very typical, in the sense that genes involved in the same function are clustered together featuring the typical modularity of phage genomes (Figure 3). These include units involved in DNA replication and repair, regulation, virion structure, and assembly (capsid, tail, and tail fibres), DNA packaging, and lysis. Further inspection of the phage genomes revealed that Langgrundblatt1, Langgrundblatt2, and Pfeifenkraut encode MazG nucleotide pyrophosphohydrolases [52], which, potentially, interfere with the hosts’ programmed cell death. The mechanism of abortive infection is widespread among bacteria and represents a typical antiviral strategy protecting the entire population by sacrificing a single cell [53,54]. Interestingly, phage Pfeifenkraut carries a group one Intron which have a limited distribution among bacteria but have been reported only in a few cases for phages infecting gram-negative bacteria [55,56]. 

With an average nucleotide identity of 97% Langgrundblatt1 and Langgrundblatt2 are members of the same species [57]. The genomes of the phages Elanor and Mallos share 75% sequence identity. Both phages contain a relatively high fraction of ORF encoding proteins of unknown functions (hypothetical proteins/CDS; 40/86 for Elanor and for Mallos 46/89) reflecting once more the significant amount of ‘dark matter’ harboured in phage genomes. 

An important family of nucleoid-associated proteins involved in the silencing of foreign (e.g., phage) DNA are H-NS proteins [58]. Interestingly, phage Laurilin encodes an H-NS-binding protein which could function as inhibitor or of the host-encoded xenogeneic silencer protein [59]. NCBI protein blast revealed a broad distribution of homolog H-NS-binding proteins among phages of gram-negative bacteria. CDD blast revealed this protein as a hypothetical protein conserved in T7-like phages (cl10202).

### 3.4. Average Nucleotide Identity (ANI) Analysis

To analyze the phylogenetic relationship of our sequenced phages, we performed a comparison based on genome wide pairwise identity. Due to the high variability within phages genomes, traditional approaches often use single-gene phylogenies for the classification; however, no single gene is shared by all seven isolated phages. Only the large terminase shares sequence identity for five of the seven phages (dark purple arrow, Figure 4). Therefore, we selected all available genomic sequences for phages infecting *Xanthomonas* reviewed by Nakayinga et al. (2021) [6] and from literature [24]. The set was further expanded by unique entries of VirusHost DB [50], as well as by our phages and their closest relatives according to NCBI nucleotide blast, resulting in a total of 97 genomes. With those genomes we performed two independent clustering analysis (using VIRIDIC and ClusterGenomes), based on average nucleotide identity (ANI). The results of the clustering dendrogram created with VIRIDIC [49] show that our isolated phages fall into four distinct clusters within the *Xanthomonas* phages (Figure 5). The ANI-based clustering analysis showed that phages Langgrundblatt1, Langgrundblatt2, and Pfeifenkraut form a distinct cluster, resulting in the novel genus “*Shirevirus*”. Phage Laurilin clusters with three phages known to infect bacterial species of the genus *Pseudomonas*. Phage Elanor clusters together with the phages Bosa, Xp12, and Xoo-sp2. Interestingly, all members of this cluster infect *Xanthomonas* species that are pathogens of plants belonging to the *Poaceae* family. Seregon clusters together with phages infecting *Xanthomonas* and *Xyllea* [24]. This is supported by both clustering approaches (Appendix A). Phage Mallos is a special case, while in VIRIDIC it clusters with the Elanor group and phage PaMx28 as a neighbour, in the Perl-based clustering it forms a group of its own. Altogether, our isolated phages display a broad diversity and cluster broadly among the known phages for *Xanthomonas*. 

This diversity within our isolates is further supported by an additional clustering performed against 700 representative phages infecting proteobacteria retrieved from VirusHost DB [50] (Appendix A).

## 4. Discussion

Phage-based products bear a great potential for the sustainable and targeted treatment of bacterial infections in agriculture. However, to the best of our knowledge, no phage-based product was registered as plant protection product or biopesticide by the European Food Safety Authority until now [61]. Some of the main challenges to move bacteriophages towards application are: gaining an comprehensive understanding of the bacteria–phage–plant interaction on a molecular level, the development of standard operation procedures for the evaluation of novel phages as pest-control agents, and reliable ‘on-field’ application strategies [61].

While many of the early phage-isolation studies [2] were limited by the technical possibilities of their time. Out of 176 known phages infecting bacteria of the genus of *Xanthomonas* only roughly 100 are sequenced. Additionally, morphological characterization has become more feasible in recent years, with more access to high-resolution imaging devices. This information provides an important basis to potentially link phage morphological traits and binding preferences known to occur in certain virion morphotypes with biocontrol possibilities [62]. Last, but not least, phages for *Xanthomonas* are highly under-sampled given the fact that the number of known phages has almost doubled in the last decade, reaching over 14,244 complete sequenced phage genomes by the beginning of 2021 [63,64,65].

Here we report the isolation and characterization of seven novel *Xanthomonas* phages. Genomic analysis revealed typical arrangement of genes into clusters linked to functional units, but also a significant amount of ‘dark matter’ harboured in phage genomes. The phages Elanor, Mallos, and Seregon encode a Cas4 family exonuclease which normally plays a role in acquiring functional spacers in bacterial CRISPR immunity [66]. Nevertheless, cases have been reported were phage-derived Cas4-like proteins led to host spacer acquisition and, subsequently, autoimmunity of the bacterial host [67]. This way, the phage uses the bacterial genome as a decoy for its own immune system, thereby gaining time for the production of phage progeny.

An essential trait for phages used in biocontrol is a lytic lifestyle. Since temperate phages could equip their bacterial host with further virulence traits [68]. All our phages were predicted to have a lytic lifestyle by the machine-learning-based phage lifestyle determination tool PhageAI [48]. This was further underpinned by the absence of integrase genes within their genomes. These results are in line with genomic clustering analysis where they cluster with virulent phages. It has, however, to be noted, that phage Seregon forms turbid plaques on lawns (Figure 1). Therefore, further analysis is required to determine the lifestyle of this phage.

Characterization of the host range assay showed that phages Langgrundblat1, Langgrundblat2, Pfeifenkraut, Elanor, and Seregon are highly specific, only infecting their isolation host and the other plant-pathogenic Xanthomonads but featuring a significantly lower efficiency of plating. Nevertheless, these assays confirmed that they do not infect the majority of the here-tested plant growth-promoting bacteria (PGPB). This shows the advantage of phages as targeted plant biocontrol argent in contrast to other antimicrobials, which also broadly affect the beneficial part of the plant microbiome. Phage Laurilin displayed the highest specificity, only infecting its isolation host *X. translucens*. In contrast, phage Mallos displayed the broadest host range and also infected the plant pathogenic bacterium *Pseudomonas syringae* DC3000, making it an omnylitic phage infecting species of two different alphaproteobacterial genera. This makes sense in the context that *Xanthomonas* and *Pseudomonas* are among the most abundant genera in the plant phyllosphere [69]. In addition, a recent study of two phages infecting *Pseudomonas syringae* pv. *tomato* DC3000 also finds lytic activity on two Xanthomonas species [70]. 

ANI-based comparison of our isolates to phages from the literature [6,24] revealed that the phages are very diverse. Phages Pfeifenkraut, Langgrundblatt1, and Laggrundblatt2 cluster with each other but show very little homology with other described phages, therefore forming the new genus of “*Shirevirus*”. This highlights that there is a vast number of uncharacterized phages out there that can potentially be harnessed for biocontrol. This is also in line with a recent phage metagenomic study of wheat phyllosphere where *Xanthomonas* and *Pseudomonas* were the most abundant bacterial genera in the plant phyllosphere and 96.8% of the generated viral taxonomic units did represent phages which have not been isolated so far [69]. 

Our results suggest that our phages are suitable candidates for an application in planta and, thereby, will add up to the many phage biocontrol trials performed against a broad spectrum of different plant pathogenic bacteria in the recent years [4,5,6,61]. A comprehensive overview over the current phage isolates for *Xanthomonas* species is given by Nakayinga et al. (2021) [6]. Further the application of Xcc phages (FoX6 and FoX2) at different stages of plant lifecycle (seed, seedling, and field condition) was accessed recently, showing reduction of disease severity at all stages demonstrating the potential benefits of phages as biocontrol agents [32].

Recent trials aim at circumventing problems caused by classical chemical treatments against Xtt, for example, by using supernatants of lactic acid bacteria, *Bacillus* strains, or plant natural products [71]. The main downside of all these substances is that they—unlike phages—cannot multiply themselves in the presence of the pathogen. Combinatorial approaches using phages and plant-growth-promoting bacteria producing those chemically active compounds could overcome this hurdle and could be part of future integrated pest control strategies to maximize crop yield and protection. Here, the phages isolated and characterized in this study add suitable candidates to be benchmarked in future biocontrol experiments for the treatment of Xtt and Xcc infections in planta.

Additionally, by way of an example from personalized medicine, the treatment of bacterial infections in humans or animals is routinely based on the isolation of bacterial strains from the patient and isolation of specific phages for the respective pathogen [72,73,74]. This approach has shown promising results and demands for the regular update of phage cocktails in the clinic. It is a strong advantage of phage-based biocontrol, that this natural diversity can be harnessed. This is very likely also required for sustainably successful agricultural applications and would require phage isolation from plants growing exposed to pathogenic bacteria in field conditions. 

## Figures and Tables

**Figure 1 viruses-14-01449-f001:**
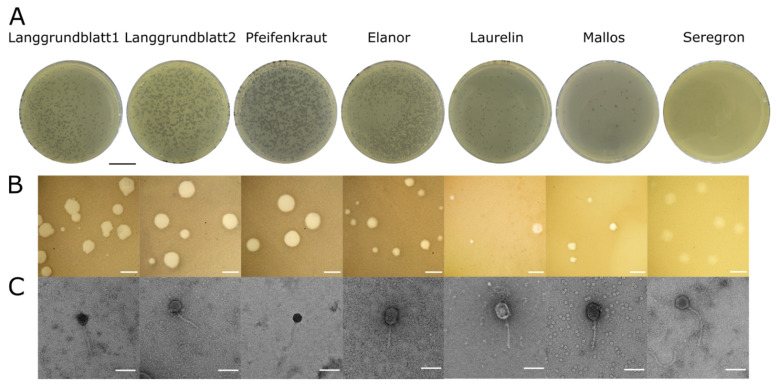
Phage morphology of novel *Xanthomonas* phage isolates. (**A**) Plaque morphologies of the seven different phages infecting *Xanthomonas translucens* pv. *translucens* (DSM 18974) (Langgrundbatt 1, Langgrundbatt 2, Pfeifenkraut, Elanor, Laurelin, and Mallos) and *Xanthomonas campestris* pv. *campestris* (Seregon). Scale Bar: 2 cm; (**B**) Stereo microscopy of single plaques. Scale bar: 1 mm; (**C**) Transmission electron microscopy (TEM) images of virion particles. The phage isolates were negative stained with uranyl acetate. Scale bar: 100 nm.

**Figure 2 viruses-14-01449-f002:**
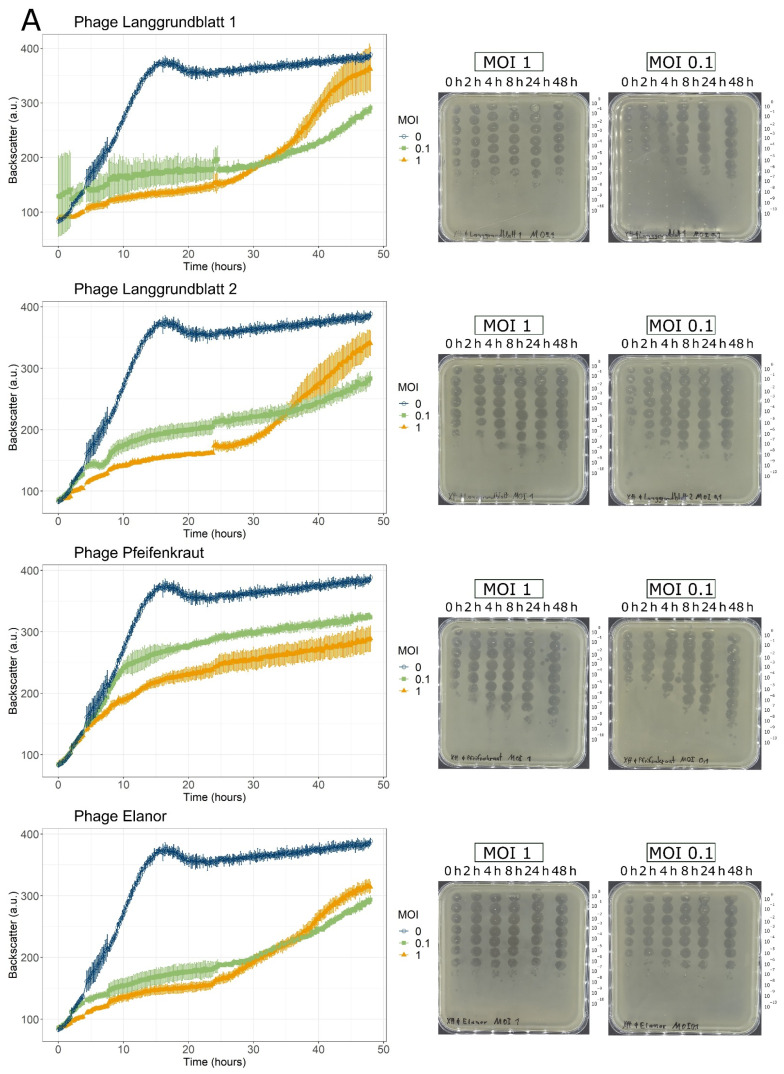
Infection curves of seven phages infecting Xtt (**A**) and Xcc (**B**). Full-grown overnight cultures of the host were adjusted to an OD_600_ of 0.2 and phages at the corresponding multiplicity of infection (MOI) were added. The infection was performed as biological triplicates. Backscatter was measured over time (left panels), as well as phage titers (right panels). Plaques of phage Seregon have a turbid appearance and are, therefore, hardly visible (see Figure 1B).

**Figure 3 viruses-14-01449-f003:**
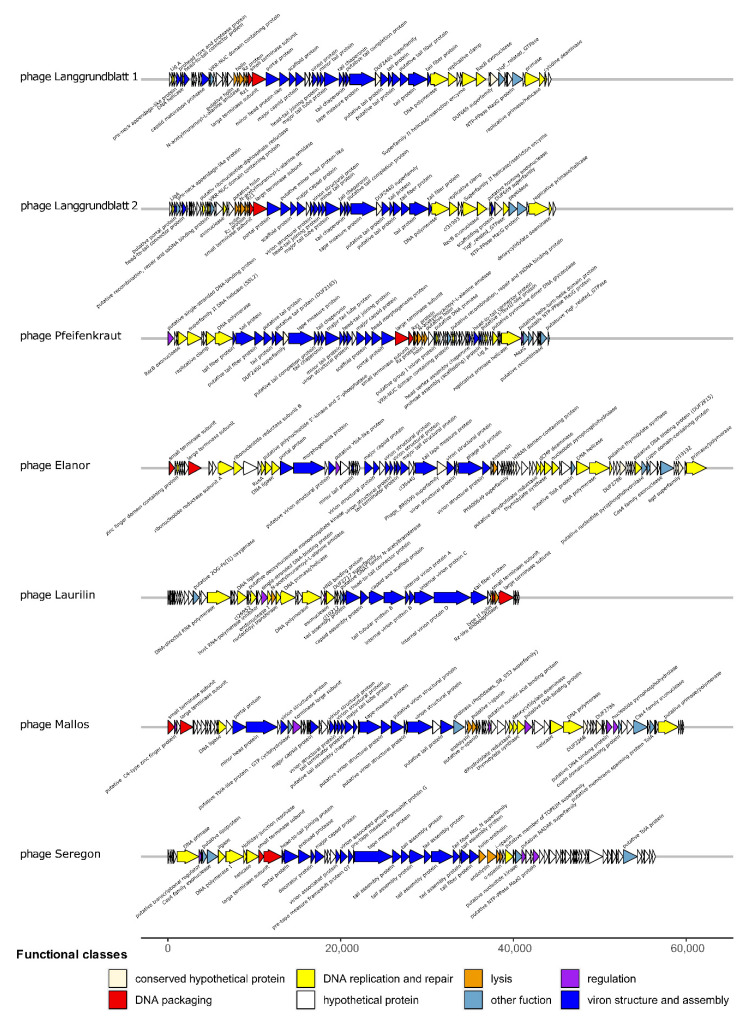
Genome map of the six *Xanthomonas* phages. Open reading frames (ORFs) were predicted using multiPhate [44], with Phanotate [42] and were later curated manually using NCBI Blast. Encoded protein domains were further predicted by using the batch function of NCBI Conserved Domain Database (CDD). Genome maps were created using the R package gggenes using fixed-length parameters.

**Figure 4 viruses-14-01449-f004:**
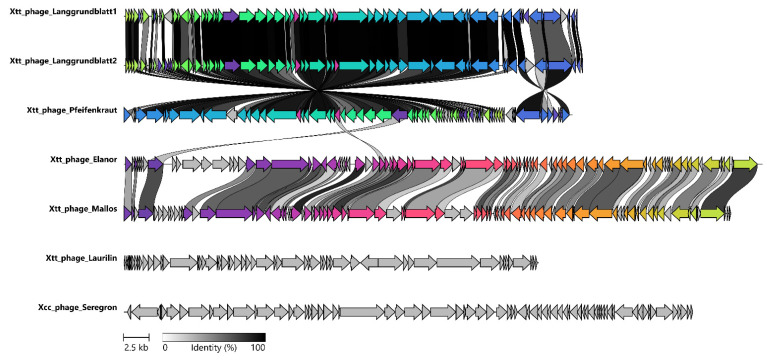
Genome comparison on the level of coding sequences. The coding sequences (CDS) of the isolated phages were compared using the clinker pipeline [60] to cluster them in groups by similarity (each colour represents one group) and per cent identity of the member of one group is indicated by grey values. The circular genomes are represented linearly and the direction of the arrows is in line with transcription direction of each CDS.

**Figure 5 viruses-14-01449-f005:**
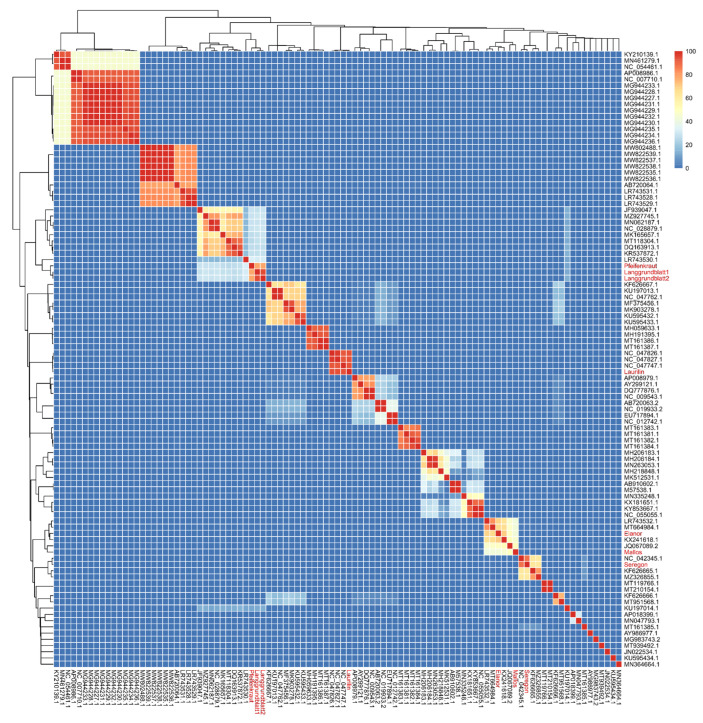
Average nucleotide-based dendrogram analysis using 97 phage genomes of phages infecting *Xanthomonas* species. Genomes are derived from [6] or Virus Host DB (https://www.genome.jp/virushostdb/, accessed on 19 January 2022). Furthermore, 11 genomes were acquired from NCBI based on similarity to our seven novel phages.

**Table 1 viruses-14-01449-t001:** Basic genomic features of the seven novel phages.

Phage Name	Accession Number	Reference Host	Genome Size (Bp)	GC Content (%)	ORF Number ^a^	Genome Termini Class ^b^	Lifestyle Prediction ^c^
Langgrundblatt 1	ON189042	*Xanthomonas translucens* DSM 18974	44.239	53.3	67	Headful (pac)	virulent
Langgrundblatt 2	ON189043	*Xanthomonas translucens* DSM 18974	44.768	53.4	68	Headful (pac)	virulent
Pfeifenkraut	ON189044	*Xanthomonas translucens* DSM 18974	43.791	53.3	72	Headful (pac)	virulent
Elanor	ON189045	*Xanthomonas translucens* DSM 18974	62.341	64.5	86	Headful (pac)	virulent
Laurelin	ON189046	*Xanthomonas translucens* DSM 18974	40.498	57.4	56	DTR (short)	virulent
Mallos	ON189047	*Xanthomonas translucens* DSM 18974	59.242	61.8	88	Headful (pac)	virulent
Seregon	ON189048	*Xanthomonas campestris*	55.527	63.2	72	COS (5′)	virulent

^a^ Open reading frames (ORFs) were predicted using multiPhate [44] with Phanotate [42] and, subsequently, annotated against different customized databases (NCBI viral proteins, NCBI Refseq proteins, Kyoto Encyclopedia of Genes and Genomes (KEGG) [43], Phage Annotation Tools and Methods (PhAnToMe) (www.phantome.org), and phage Virus Orthologous Groups (pVOG)). Additionally, manually curated using NCBI Blastp. Encoded Protein Domains were further predicted by using the batch function of NCBI Conserved Domain Database (CDD). ^b^ Genome termini classes were determined using PhageTerm [47]. ^c^ Phage lifestyle was predicted by the machine-learning-based program PhageAI [48] and, further, confirmed by absence of intergrade genes inside the genomes.

## Data Availability

Not applicable.

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
