# Peer review of "Isolation of Novel Xanthomonas Phages Infecting the Plant Pathogens X. translucens and X. campestris"

_viruses, 2022, doi:10.3390/v14071449_

Round 1
Reviewer 1 Report
Erdrich et al, isolated and characterised seven phages that infect Xantamonsa translucens and Xanthomonas campestris. The phages were collected from wastewater and soil and characterised by electron microscopy and genome sequencing. Killing curve assay and host range assay were also performed. I have some major and minor points that the authors should consider:
1) Introduction is poorly written, it jumps from one place to another without a clear direction. Maybe first start by telling what the problem is and then how phage can be used to solve the problem.
2) Material and methods isnot written sufficiently. It should be re-written in details on what has been done, so the exp can be repeated by a reader.
3) Results is a mix of results and discussion.
4) Discussion needs more referencing. Wat about other phages that infect other pathogens. There is no such mention at all. Discussion suddenly and unexpectedly ends with no clear conclusion.
L38-39: provide a reference, which phages and against which Xanthomonas pathovars? This sentence needs rephrasing as well.
L41: what are the “trans-41 lucens group”?
L51: is this the same colony forming morphology as Xtt?
L55: move this ‘but there is still a great need for further phage isolates applicable for biocontrol’ to after the phage examples.
L58: explain why chronic infections hampers biocontrol application?
L61: desirable traits? What are they?
L64: these are not desirable traits! These are to better characterise and understand the phages, so that we can decide if we can use them as biocontrol agents.
L68: is this Xtt?
L70: is it Xcc?
L70-72: instead of this sentence that belongs to the result section, summarise how your finding will contribute to the phage biocontrol section!
L77-78: rephrase this sentence, maybe to: Nutrient broth or agar (NB or NBA) was used for culturing the bacterial strains at 30C.
L86: remove the second buffer, PBS is phosphate-buffered saline
L87: what is a rock shaker? As what speed?
L91: be specific, what is a fully-grown overnight culture of the host? What is cfu per ml?
L94-96: again what is a fully-grown overnight culture of the host? What is 1X and 5X NB?
L99: donot need to repeat (Sarstedt; Filtropur S, PES).
L102: re streaking phage? How? Do you mean plaque assay?
L105: after overnight incubation of what?
L105-108: coulnot you just take a single plaque to purify?
L106: what is SM-buffer?
L106: rock shaker?
L113: change Pfu/mL to PFU per ml or PFU mL-1
L120: change 3,5 to 3.5
L129: 1200 rpm? Excessive! Or Is it 200 rpm? Shaking constantly?
L131: OD600 of 0.2 is equal to ?? CFU per ml.
L132 what is MOI?
L134: what is RT?
L137: change our to the and throughout the ms.
L142: what is LB?
L149: free DNA! do you mean chromosomal DNA?
L213: but with considerable variation in plaque sizes.? What ddoes this eman?
Figure 2: what is MOI of 0? Do you menan Xtt or Xcc with no phage?
Figure2: phages are resisting to the bactreia quite quickly. perhaps best to use difefrent MOI?
L251: it isnot clear if the spot assay shown in Figure to comes from the killing curve assay? Does it mean the plate reader was stopped every few hours so you could take samples?
L259: what is a slight susceptibility?
L261: what is at low levels.?
L262: what is EOPs?
L266: this sentence belongs to discussion.
L313-218: belongs to discussion.
L320: change she to the.
L326: too many Interestingly used in result section. Remove.
Figure 4: poor quality, hard to read.
L369: what about after 2020?
L373: this sentence is hard to understand. Rephrase.
L399: what is EOP?
L399: this statement is false. Above you said phage Mallos showed additional lytic ac-262 tivity on the plant pathogenic P. syringae strain DC3000, but also infected the plant 263 growth-promoting P. fluorescens. So one the phages infect a benfecial bacterium.
Discussion should finish by making a statement of why this research was essential, but it rather finishes suddenly and unexpectedly.
Author Response
Comments to Author from Referees (Manuscript ID: viruses-1754070):
Referee: 1
Erdrich et al, isolated and characterised seven phages that infect Xantamonsa translucens and Xanthomonas campestris. The phages were collected from wastewater and soil and characterised by electron microscopy and genome sequencing. Killing curve assay and host range assay were also performed. I have some major and minor points that the authors should consider:
- Introduction is poorly written, it jumps from one place to another without a clear direction. Maybe first start by telling what the problem is and then how phage can be used to solve the problem.
A: The major problem here was probably that we introduced two pathogens one after the other (Xtt and Xcc). We now tried to make this more clear and also shortly introduced the ‘bigger picture’ in the beginning. We, however, also find it important to provide a concise introduction into the aspects relevant for this study.
- Material and methods is not written sufficiently. It should be re-written in details on what has been done, so the exp can be repeated by a reader.
A: Thank you for this comment. We agree with the reviewer that this is important and that many procedures were not properly described. We now provied better descriptions and appropriate references in the revised version.
- Results is a mix of results and discussion.
A: We moved some aspects (potential gene functions of the phage isolates) to the discussion.
- Discussion needs more referencing. Wat about other phages that infect other pathogens. There is no such mention at all. Discussion suddenly and unexpectedly ends with no clear conclusion.
A: An overview over the field of phage biocontrol and comprehensive reviews about phages infecting other pathogenic bacteria is given in: Buttimer et al. 2017, Svircev et al. 2018, Nakayinga et al 2021, Holtapples et al 2021, which are clearly cited in the paper.
The vast majority of the achievements in the field is out of the scope of this study focusing on isolation of phages for two plant pathogenic members of the Genus of Xanthomonas. Never the less we included an short overview of the field:
“An overview of the economically most relevant plant pathogens is given in Mansfield et al 2012 [7]. In the resent years, several attempts have been made to isolate phages infecting plant some of those pathogens, including Ralstonia solanacearum, Erwinia amylovora, Pseudomonas syringe spp, Xylella fastidiosa, and Xanthomonas spp.. Multiple studies were conducted on phage biocontrol of Ralstonia solanacearum showing suppression of plant wilting on potato and tomato [8–13]. Also Erwinia amylovora, which developed resistance to streptomycin has led to evaluation of phage biocontrol with promising outcomes in some cases [14–16]. Due to the broad spectrum of plants infected by Pseudomonas syringe spp., multiple biocontrol trails have been conducted in the past, but with a special effort on citrus cranker disease [17–23]. Xylella fastidiosa is a mayor threat to olive trees in Europe, here first phages have been isolated and tested [24–26]. Due to the economical relevance of Xanthomonas spp., a growing body of phage isolates is available for this genus.
”
L38-39: provide a reference, which phages and against which Xanthomonas pathovars? This sentence needs rephrasing as well.
A: Changed to: “For other Xanthomonas species [6], first phages have been sequenced and morphologically characterized, but in the case of Xtt no phages are available in phage collections.”
L41: what are the “trans-41 lucens group”?
A: The bacterial species of Xanthomas translucens contains multiple pathovars, namely: pv. undulosa, pv. translucens, pv. cerealis, pv. Hordei and pv. Secalis. They altogether are referred as the “translucens group” [27]
L51: is this the same colony forming morphology as Xtt?
A: No. Both bacteria form yellow colonies, the genus got its name from the Greek word for yellow – Xanthos. But Xanthomonas translucens (Xtt) forms slightly more transparent or translucent colonies.
L55: move this ‘but there is still a great need for further phage isolates applicable for biocontrol’ to after the phage examples.
A: Done
L58: explain why chronic infections hampers biocontrol application?
A: Phages with a chronic lifestyle use the bacterium for production of new phage particles and have a fitness cost for the infected bacterium, but do not kill it during infection. In contrast to lytic phages, were a phage infects the bacterium, uses its resources for replication and then kill the host bacterium. For biocontrol of pathogenic bacteria, the latter is preferred since the aim is to irradiate the harmful bacterium. This is now further explained in the introduction as well, see lines 76-79.
L61: desirable traits? What are they?
A: This is indicated at the end of the paragraph. Desirable traits include a lytic live style, sequenced genome, repression of host growth at low multiplicity of infection (MOI), stable production of virion progeny, known host range, good storage properties etc.. See line 81.
L64: these are not desirable traits! These are to better characterise and understand the phages, so that we can decide if we can use them as biocontrol agents.
A: Correct. This was a bit misleading. We have re-worked this paragraph:
“One of the most important aspects of phage biocontrol is the isolation of phages with desirable traits, including a lytic life style, the absence of virulence genes and reasonable repression of host growth. Comprehensive characterization, genome analysis and the analysis of the host range of newly isolated phages allow us to better understand the ways how phages target their host bacterium. Altogether, these efforts are important to expand the “toolbox” for sustainable and targeted control of plant pathogens and allows researchers to select phages with most desirable traits.”
L68: is this Xtt?
A: Yes. Changed as requested.
L70: is it Xcc?
A: Yes. Changed as requested
L70-72: instead of this sentence that belongs to the result section, summarise how your finding will contribute to the phage biocontrol section!
A: It is not unusual that the last paragraph of the introduction already provides a teaser on the results to be described. We briefly mention that the phages “…have a lytic live style making them suitable candidates for phage biocontrol.”
We would therefore prefer to leave it as is. It would be too speculative to dive further in potential application of the phages as we focus on the basic characterization in this study. Ongoing efforts focus on biocontrol experiments in planta.
L77-78: rephrase this sentence, maybe to: Nutrient broth or agar (NB or NBA) was used for culturing the bacterial strains at 30C.
A: We rephrase this sentence, but to reasons of clarity, we would prefer “NB agar” over “NBA”
L86: remove the second buffer, PBS is phosphate-buffered saline
A: Thanks, this was corrected.
L87: what is a rock shaker? As what speed?
A: A rocking shaker combines the ability of a rocker and a shaker it combines wiping and rotation movement. We added the information on the speed (30 rpm).
L91: be specific, what is a fully-grown overnight culture of the host? What is cfu per ml?
A: A densely grown culture of Xtt or Xcc is typically OD600 of 1. This information was added in the manuscript.
L94-96: again what is a fully-grown overnight culture of the host? What is 1X and 5X NB?
A: We have re-worked this part. This was probably misleading. The incubation was performed in 1x NB (by using a concentrate, 5x).
“ Phage enrichment was performd in 15 ml NB. Filtered supernatant solution (1mL) and 1 mL of an overnight culture of the host was added. To adjust the enrichment culture to 1-fold NB, appropriate amounts of 5-fold concentrated NB were added to the sample. The culture was incubated at 30°C and at 150 rpm overnight.”
L99: donot need to repeat (Sarstedt; Filtropur S, PES).
A: Done
L102: re streaking phage? How? Do you mean plaque assay?
A: Yes, this is a microbiological standard procedure. By dipping in the plaque with an inoculation loop and restreaking onto a fresh double agar overlay containing the host bacterium one can generate new single plaques.
“Purification of the phage samples was carried out by re-streaking single plaques with an inoculation loop on a fresh double agar overlay at least 3 times. When a stable plaque morphology was observed after three re-streakings, a sample was considered as a single phage isolate [16].”
L105: after overnight incubation of what?
A: Now clarified in the text, line 132. Overnight incubation of double agar overlay containing the purified phage.
L105-108: coulnot you just take a single plaque to purify?
This is indeed what happens when the phage gets resrteaked 3 times on a double agar overlay. Lines 105-108 deal with the procedure of propagation of a purified phage originated from a single plaque. This procedure is indeed important, since single plaques after the enrichment culture may still be composed of different virion types.
L106: what is SM-buffer?
A: thanks for the comment. This is referring to salt-magnesium buffer in literature only referred as SM. It is a standard storage buffer for phage particles; the recipe is added, line 134.
L106: rock shaker?
A: we believe that is a common term, see comment above.
L113: change Pfu/mL to PFU per ml or PFU mL-1
A: Done
L120: change 3,5 to 3.5
A: Done
L129: 1200 rpm? Excessive! Or Is it 200 rpm? Shaking constantly?
A: This is a Microtiter plate cultivation system called BioLector® to ensure oxygen supply in small volumes the manufacturer recommends this speed and the bacteria grow really well under these conditions. So, yes, this is correct.
L131: OD600 of 0.2 is equal to ?? CFU per ml.
A: 5.1*108 CFU per ml
L132 what is MOI?
multiplicity of infection (MOI). It describes the ratio of infective particles compared to entities serving as potential host. Now added in the text, line 162-163.
L134: what is RT?
A: Room temperature – added now
L137: change our to the and throughout the ms.
A: Changed as requested
L142: what is LB?
A: Lysogeny Broth – added now
L149: free DNA! do you mean chromosomal DNA?
A: Here specifically, all kinds of not enclosed DNA are addressed. You are right, reminiscent of bacterial chromosomal DNA generated during lysis are a part of it, but also non-intact phage particles can contribute to not enclosed DNA in the solution. Since for sequencing only intact phage genomes are desirable, all kinds of free DNA were removed in this step.
L213: but with considerable variation in plaque sizes.? What does this eman?
A: Some phages like the model phage T7 have variable plaque sizes. This means that the outcome of the first round of infection is crucial for subsequent plaque formation. Since bacteria have multiple defence systems in place, a partial activation could lead to lower phage progeny in the first round of infection, but also nutrient status of the bacteria being infected is determining how many phage particles can be formed. So, it’s a complex multifactorial interaction. The standard deviation is 0.2 mm for phages Langgrundblatt1 and Langgrundblatt2.
Figure 2: what is MOI of 0? Do you menan Xtt or Xcc with no phage?
A: Yes, this is the control with no phage.
Figure2: phages are resisting to the bactreia quite quickly. perhaps best to use difefrent MOI?
A: Different MOIs were tested, further testing of other MOIs would not substantially contribute to the findings. 50h of cultivation is already quite long for a 1 mL culture. Ongoing experiments now focus on the application of the phages in planta. Here, conditions differ considerably and therefore optimizing the in vitro conditions in microtiter plates should not be over interpreted.
L251: it isnot clear if the spot assay shown in Figure to comes from the killing curve assay? Does it mean the plate reader was stopped every few hours so you could take samples?
A: It’s not a plate reader, its a Microtiter plate cultivation system called BioLector®. Sampling occurred under sterile conditions in-between the measurements.
L259: what is a slight susceptibility?
A: The EOP values are provided in Table S2, as referred in the text.
L261: what is at low levels.?
A: The EOP values are provided in Table S2, as referred in the text.
L262: what is EOPs?
Efficiency of plating - EOP. Described in L176-177, is the relative infection effectivity of a phage on another bacterium in comparison to the bacteria it normally propagates on or on which it was isolated from the environment, also called “host”.
L266: this sentence belongs to discussion.
A: Moved to the discussion
L313-218: belongs to discussion.
A: moved to discussion
L320: change she to the.
A: Done
L326: too many Interestingly used in result section. Remove.
A: Done
Figure 4: poor quality, hard to read.
A: Side effect of putting it into MS Word. Of course, we provide Figure 4 in high quality in the final upload
L369: what about after 2020?
A: Changed as requested. Also checked European Food Safety Authority (EFSA) again today, no approval for a phage-based plant biocontrol product was approved yet. One application from APS Biocontrol Ltd. (UK) - EFSA-Q-2021-00670
L373: this sentence is hard to understand. Rephrase.
A: Changed to: “While many of the early phage isolation studies [2] were limited by the technical possibilities of their time”
L399: what is EOP?
A: Efficiency of plating EOP. Described in L176-177, is the relative infection effectivity of a phage on another bacterium in comparison to the bacteria it normally propagates on or on which it was isolated from the environment also called “host”.
L399: this statement is false. Above you said phage Mallos showed additional lytic ac-262 tivity on the plant pathogenic P. syringae strain DC3000, but also infected the plant 263 growth-promoting P. fluorescens. So one the phages infect a benfecial bacterium.
A: yes, but only very weakly. However, this fact is mentioned in the results and discussion.
Discussion should finish by making a statement of why this research was essential, but it rather finishes suddenly and unexpectedly.
A: Since this manuscript is focusing on the isolation and characterization of the phages – too much speculation on application which was not tested yet, should be avoided. However, we tried to round it up a bit better.
“With the phages isolated and characterized in this study, we add suitable candidates to be benchmarked in future biocontrol experiments for the treatment of Xtt and Xcc infections in planta”
Reviewer 2 Report
The research article by Edrich et al. about the Xanthomonas phages is well-designed and reported. The authors have made a tremendous effort to present the perfect article in its present form.
1. The major question is about the applicability of these phages to the plants. What is the author's experience with the application of phages against plant pathogens on the plant? Instead of the mere phage characterisation report, it will be useful for the readers if it is discussed.
2. Another point is the bacterial strains used in this study because phages are specific to a species or strain of bacteria so these phages may not be suitable across the strains of Xanthomonas. Maybe the authors should have isolated bacteria from the plant samples for this study, at least from a particular location.
Author Response
Referee: 2
The research article by Erdrich et al. about the Xanthomonas phages is well-designed and reported. The authors have made a tremendous effort to present the perfect article in its present form.
- The major question is about the applicability of these phages to the plants. What is the author's experience with the application of phages against plant pathogens on the plant? Instead of the mere phage characterisation report, it will be useful for the readers if it is discussed.
A: The application of these phages against plant pathogens is a focus of ongoing studies. We are in the process of establishing meaningful infection experiments in planta. But here further experiments are required and we would prefer to avoid too much speculations in the discussion. Others have done much in the area of plant biocontrol:
“]. In the resent years, several attempts have been made to isolate phages infecting plant pathogens, including Ralstonia solanacearum, Erwinia amylovora, Pseudomonas syringe spp, Xylella fastidiosa, and Xanthomonas spp.. Multiple studies were conducted on phage biocontrol of Ralstonia solanacearum showing suppression of plant wilting on potato and tomato [8–13]. Also Erwinia amylovora, which developed resistance to streptomycin has led to the evaluation of phage biocontrol with promising outcomes in some cases [14–16]. Due to the broad spectrum of plants infected by Pseudomonas syringe spp., multiple biocontrol trails have been conducted in the past, but with a special effort on citrus cranker disease [17–23]. Xylella fastidiosa is a mayor threat to olive trees in Europe, here first phages have been isolated and tested [24–26].”
- Another point is the bacterial strains used in this study because phages are specific to a species or strain of bacteria so these phages may not be suitable across the strains of Xanthomonas. Maybe the authors should have isolated bacteria from the plant samples for this study, at least from a particular location.
A: Very good point and this will definitely be tackled in future efforts. Especially for phage therapy in humans personalized medicine is based on the isolation of bacterial strains from the patient and isolating specific phages or testing phages collections against those bacterial strains has shown promising results. Also, in the field of agriculture this is probably the best practice in the future. At the moment, despite all recent developments, the field is not quite there yet. Big phage collections for plant pathogenic bacteria are lacking and field application is still experimental in many countries, but for future application-based collaborations with farmers we definitely would consider this approach.
We included this aspect in the discussion:
“Additionally, by way of an example from personalized medicine, the treatment of bacterial infections in humans or animals is routinely based on the isolation of bacterial strains from the patient and isolation of specific phages for the respective pathogen [72–74]. This approach has shown promising results and demands for the regular update of phage cocktails in the clinic. It is a strong advantage of phage-based biocontrol, that this natural diversity can be harnessed. This is very likely also required for sustainably successful agricultural applications and would require phage isolation from plants growing exposed to pathogenic bacteria in field conditions. “
Round 2
Reviewer 1 Report
The authors responded to the comments sufficiently.